# Massively Multi-Agents Role Playing: Simulating Financial Market Dynamics with LLMs

## Abstract

Large Language Models (LLMs) have been trained on vast corpora of data, allowing them to learn internal representations of how humans would respond in different scenarios. This makes them well-suited to simulate the actions of market participants, to model their collective impact on financial markets and perform financial forecasting. However, there also exist various sources of errors that could affect the effectiveness of LLM agent-based simulations of the market. Firstly, individual market participants do not always make rational decisions, which might not be captured by the logical reasoning process of LLMs. Secondly, the numerical and financial literacy of LLMs are also not highly reliable, due to possible knowledge gaps in their numerical understanding and possible hallucinations in their outputs. To tackle these issues, we propose our Massively Multi-Agents Role Playing (MMARP) method, which aims to produce highly accurate market simulations through theory-driven prompt designs. To reduce the impact of noisy actions caused by individual irrational investors, we leverage the LLM-generated next-token weights to simulate repetitive prompting, and obtain the aggregated market response. To minimize the effects of possible gaps in its numerical knowledge or potential hallucinated outputs, we prompt the LLM using a range of price inputs for each trading day. Finally, to produce simulated forecasts of market prices, we perform the above prompting strategies across two types of LLM-agent roles, buyers and sellers, and obtain the intersection price between their response curves. Through experimental results, we show that MMARP can outperform other deep-learning methods and various financial LLMs in forecasting metrics.

## 1 Introduction

Financial markets are complex ecosystems that are driven by millions of market participants, each making individual decisions about the value of an asset based on available information (Fama, 1970). While traditional deep-learning models have been developed to predict the market in the past (Ding et al., 2015; Hu et al., 2018; Xu & Cohen, 2018), they typically do so by identifying the historical patterns in market data, but do not fundamentally capture the individual decision-making processes that drive these patterns. On the other hand, Large Language Models (LLMs), which have been trained on vast corpora of human-produced data, have demonstrated the ability to learn internal representations (Allen-Zhu & Li, 2023; Chen et al., 2024) of how humans might respond to different prompts, enabling them to simulate human decision-making. This raises the possibility of simulating the actions of market participants using LLM agents, to model their collective impact on the market.

Generally, works that utilize LLM agents in Finance (Zhang et al., 2024b; Yu et al., 2024a;b) have focused on using them in advisor roles to enhance investor decision-making. For these works, the goal is typically to maximize profits, and these models are only evaluated over their profitability metrics (*e.g.,* cumulative returns, Sharpe ratio). In contrast, our work seeks to use LLM agents to simulate investor actions to study actual market dynamics, which could offer a novel framework for financial researchers to understand and test hypotheses about market behavior. The accuracy of our simulation would also be evaluated over forecasting metrics. This has been previously explored in LLM agent-based simulation works, such as modeling pandemic spread across a population (Chopra et al., 2024), or the U.S. election results (Zhang et al., 2024c). In the Finance domain, this has not been studied in detail. Some recent works have began to explore the use of LLM agents to model investor actions (Zhang et al., 2024a; Gao et al., 2024), but these were evaluated qualitatively based

on how reasonable their behavior is, but not their actual predictive performances. In this work, we aim to produce effective simulations of actual market behavior to generate accurate price predictions.

However, accurately simulating participants in the market is a difficult task. We can identify two challenges: Firstly, individual market participants do not always make rational decisions (Daniel & Titman, 1999). LLMs, while excelling at performing logical reasoning, might not be able to capture these irrational behavior when simulating investor actions (Alsagheer et al., 2024; Ma et al., 2024), which reduces their effectiveness in this use case. Secondly, the numerical and financial literacy of LLMs are also not highly reliable. In the past, LLMs have been shown to make simple but crucial mistakes when handling numerical values[1], or produce hallucinations when performing reasoning on financial texts (Koa et al., 2024). Because of these limitations, it might not be fully reasonable to assume LLM agents can accurately replicate actual investor behavior (see Figure 1), which would reduce the effectiveness of utilizing LLM agents to model participant behaviors in financial markets.

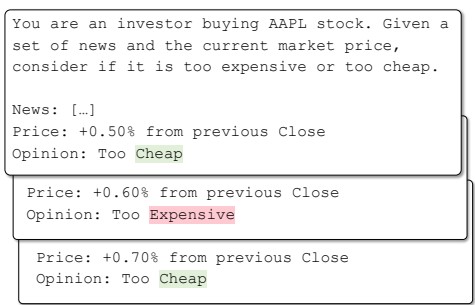 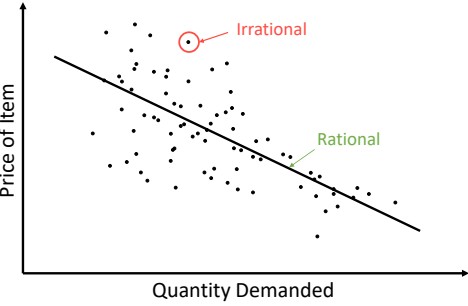

Figure 1: Left: Given the *same* set of news and a range of prices, the LLM does not give consistent judgments on whether the price is too "cheap" or "expensive". Right: Market participants do not always make rational decisions, resulting in actions that are not the most optimal. Because of these, it might not be reasonable to assume LLM agents can accurately replicate actual investor behavior.

To deal with the above-mentioned challenges, we propose our Massively Multi-Agents Role Playing (MMARP) framework, which utilizes a series of theory-informed prompt designs to produce highly accurate market simulations. Firstly, to reduce the impact of noisy actions caused by individual irrational investors, we leverage LLM-generated next-token weights to simulate repetitive prompting in order to obtain the aggregated market response, which is known to be less noisy and observable. Secondly, to minimize the effects of possible gaps in its numerical knowledge or potential hallucinated outputs, we prompt the LLM across a range of price inputs for each trading day to obtain the aggregated response function. Finally, to produce simulated prediction of market prices, we perform the above prompting strategies across two types of LLM-agent roles, buyers and sellers, and obtain the intersection price between their response functions. Crucially, this is also the same mechanism in which market equilibrium prices are determined in economic markets (Mankiw & Taylor, 2020).

To demonstrate the effectiveness of MMARP, we perform experiments over some financial datasets and show that our method can outperform other deep-learning methods and various financial LLMs in forecasting metrics. In addition, we do a rigorous model study to show that the simulated behavior is valid, by comparing our generated response curves to theory-based demand curves in economics.

The main contributions of this paper can be summarized as:

- We investigate the validity of using LLM agents to model market participants in financial markets, which has not been extensively studied before this work. We observe various sources of stochastic errors that could reduce the overall simulation accuracy, which include irrational investor behaviors, the lack of numerical understanding in LLMs, and possible hallucination in LLM outputs.

- We propose a method that aims to produce highly accurate market simulations through theory-driven prompt designs. This is done through simulating the aggregate market response using the LLM-generated next-token weights, prompting across a range of prices for each trading day, and using the intersection point between the LLM response curves to obtain accurate price forecasts.

---

[1]LLMs are known to make simple but crucial mistakes when handling numbers, such as comparing the magnitude between 9.11 and 9.9. See: https://x.com/goodside/status/1812977352085020680.

- We conduct experiments across multiple financial datasets to show that MMARP can outperform other LLM-agent based methods in both the forecasting and profitability metrics. We also perform an extensive model study to verify the validity of the simulated market behavior using MMARP.

## 2 BACKGROUND

**LLM Agents in Financial Simulations.** Since the advent of Large Language Models (LLMs), works have also started to explore the use of LLM agents in Finance. Early works first explored augmented single-LLM agent models with tool-use (Zhang et al., 2024b) or memory (Yu et al., 2024a) to enhance their capabilities in making investing decisions. Later works would explore the use of multiple LLM agents, such as a group of seven analyst agents working with a manager agent (Yu et al., 2024b) to provide investing recommendations. Across these works, the main goal is generally to maximize profits, and the performances are only evaluated on their profitability metrics.

More recently, works have began exploring the use of LLM agents to simulate market participants, which are more closely related to our work. Some of these include Agent-based Simulated Financial Market (Gao et al., 2024), which simulates the actions of four different types of investors, and StockAgent (Zhang et al., 2024a), which modeled the actions of up to 200 LLM agents to study their aggregate behaviors on price trends. In these works, each LLM agent is used to represent a single investor, which could limit the effectiveness of the simulation, given that the market consists of transactions from millions of investors each day. Currently, these agent-based simulations are only studied empirically to observe if the simulated actions and overall price trends are reasonable.

Another closely related set of works are those on macro-level LLM-agent simulations. These works seek to simulate the behavior of entire systems such as the economy (Li et al., 2024), which usually consists of millions of humans and cannot be individually modeled by a single LLM agent. To do so, these works usually use a single LLM agent to model entire groups of the same archetypes (Chopra et al., 2024). For example, to predict the U.S. election results (Zhang et al., 2024c), it might be sufficient to simulate by the unique voter demographics, instead of each voter individually. Our work follows this idea by seeking to model the buyers and sellers in financial markets as a whole.

**Numerical and Financial Literacy of LLMs.** The numerical understanding of LLMs is not well-studied in literature. Different LLMs have different methods of tokenizing numbers (Jun, 2024), which could affect their numeracy level. Empirically, LLMs have been shown to fail at simple numerical tasks[1]. The root of this problem likely stems from the continuous nature of numerical values (Golkar et al., 2023). Unlike individual words, which are finite in nature, it is impossible for LLMs to encounter all possible numerical values during training, resulting in possible gaps in its numerical knowledge. It has been shown that the ability of LLMs to handle numbers correlates with how frequently those numbers occur in the train data (Razeghi et al., 2022), and they are usually unable to extrapolate outside the range of numbers they have been trained on (Wallace et al., 2019).

On the other hand, LLMs have been extensively shown to be able to process text-based financial data, through multiple tasks such as sentiment analysis and financial forecasting (Xie et al., 2023; Wu et al., 2023). However, it has also been observed that they can produce hallucinations (Koa et al., 2024) when performing reasoning on financial texts, which could reduce their overall reliability.

**Irrational Participants in Financial Markets.** In financial markets, the individual demand (Friedman, 1949) for an asset can be modeled as a function $d(X)$, where $X$ consists of the input factors such as the price or the non-price determinants such as its future price expectations (Mankiw & Taylor, 2020). For each individual, the demand function can further be split into two components:

$$d(X) = d_{rational}(X) + \epsilon(X), \tag{1}$$

where $d_{rational}(X)$ represents the non-stochastic, rational component which is typically representative of the whole market, while $\epsilon(X)$ represents the stochastic, irrational component, which can be affected by the idiosyncrasies of each individual (McFadden, 1972) and is difficult to predict.

On the market-level, the aggregate demand for an asset from all individuals can then be modeled by:

$$D(X) = \sum_{i=1}^{N} (d_{rational,i}(X) + \epsilon_i(X)), \tag{2}$$

where $N$ represents the total number of participants that are currently trading the asset in the market, and $d_{rational}(X)$ is assumed to be consistent across all individuals, when given the same inputs $X$.

Using the law of large numbers, the irrational component from each individual would have a very small impact on the overall market demand (Lal, 1975; Lux & Marchesi, 1999). Hence, we have:

$$\lim_{N \to \infty} \sum_{i=1}^{N} \epsilon_i(X) = 0. \tag{3}$$

As such, the demand for an asset in a market with a large number of participants can be modeled by:

$$D(X) \approx \sum_{i=1}^{N} d_{rational,i}(X). \tag{4}$$

This can typically be visualized as a demand curve, which is a downward sloping curve, where the gradient is determined by the price elasticity of the asset and the intercept is determined by its non-price determinants. Similarly, the supply of an asset can also be calculated the same way, resulting in a supply curve with an upward slope. Finally, the market price of an asset is typically determined by finding the intersection point between the demand and supply curves (Mankiw & Taylor, 2020).

## 3 MASSIVELY MULTI-AGENTS ROLE-PLAYING

The Massively Multi-Agents Role-Playing (MMARP) framework consists of three components. To reduce noise from individual irrational investors, we use LLM-generated next-token weights to simulate repetitive prompting, yielding an aggregated market response. To address gaps in LLMs' numerical knowledge or potential hallucinations, we prompt across a range of price inputs for each trading day to derive an aggregated LLM response function. Finally, we repeat these across buyer and seller roles to obtain the intersection of their response functions as the simulated market prices.

### 3.1 PROBLEM FORMULATION

To investigate the validity of using Large Language Model (LLM) agents to model market participants, we first look at the process in which the agents are utilized in financial multi-agent systems.

Given some input information $X$ of an asset, such as its price or news information, an LLM agent is typically prompted to produce an actionable response $\alpha$. This response may take the form of a binary buy or sell decision (Gao et al., 2024; Yu et al., 2024b) or some quantitative values indicating the price and quantity of the asset to transact (Zhang et al., 2024a). For an LLM, the response is sampled from its generated next-token probability based on the provided inputs. For example, given the price of an asset, we can prompt an LLM whether to buy an asset. This can be formulated as:

$$\alpha \sim l_d(X), \tag{5}$$

where $l_d(X)$ represents the LLM generative function that outputs the token probability for whether to buy an asset. For an LLM agent to realistically simulate a market buyer, this function should accurately emulate the demand function of the individual, *i.e.,* $l_d(X) = d(X)$, matching their actions.

### 3.2 IRRATIONAL PARTICIPANTS IN FINANCIAL MARKETS

Following Equation 1, the individual demand function $d(X)$ is composed of a rational component $d_{rational}(X)$ and a stochastic error term $\epsilon(X)$, which accounts for irrational or unpredictable individual behaviors. Given that LLMs predominantly generate outputs that are grounded in reasoning and logical patterns derived from training data (Kojima et al., 2022), they might struggle to simulate the irrational components of investor behavior accurately (Alsagheer et al., 2024; Ma et al., 2024).

To deal with this problem, a possible solution is then to generate a large number of outputs from the LLM. By drawing a large number of response samples $\alpha$, we would then be simulating the aggregate behavior across a large number of market participants, which would give us $\sum_{i=1}^{N} \alpha_i = D(X)$. Then, by following the approximation found in Equation 4, we can obtain the equivalent simulation:

$$\sum_{i=1}^{N} \alpha_i \approx \sum_{i=1}^{N} d_{rational,i}(X). \tag{6}$$

Given that LLMs excel at rational reasoning, this becomes a more realistic task, as the *stochastic* error term $\epsilon(X)$ is now minimized. This shows that while it is difficult to use LLM agents to direct model individual participants, it is possible to utilize them to simulate the aggregate market demand.

Following the assumption that each rational participant would make the same reasonable response given the same inputs, the samples can all be drawn from the same LLM prompt $l_d(X)$. In this case, by providing the LLM with a set of defined options (*e.g., Expensive, Cheap*), we can also directly extract the ratio of their generated token probabilities to represent the ratio of the expected responses from the market. This "trick" allows us to simulate the behavioral patterns of large-scale populations using LLMs without repetitive prompting, which would minimize unnecessary computation costs.

### 3.3 HALLUCINATIONS AND KNOWLEDGE GAPS IN LLMS

Next, for an LLM to accurately simulate market reasoning, it would also need to understand and respond to the input information $X$ in the same way as participants do. However, various sources of errors exist. Firstly, LLMs are known to produce hallucinations in their outputs (Zhang et al., 2023), which would result in responses that differ from actual participants. While this is a problem common to all LLM works, it has also been observed in those dealing with financial texts (Koa et al., 2024). Secondly, the ability of LLMs to understand the *numerical* price values is not well-studied in literature. Empirically, LLMs have been shown to be weak at handling numerical-based tasks (Dziri et al., 2024; Frieder et al., 2024), without the use of external tools like interpreters (Gao et al., 2023).

From the observations above, we can further break down the LLM generative function into two:

$$l_d(X) = \bar{l}_d(X) + \bar{\epsilon}(X), \tag{7}$$

where $\bar{l}_d(X)$ is the general internal knowledge representation of the LLM, while $\bar{\epsilon}(X)$ represents the stochastic error term that can arise from both its imprecise numerical understanding and the hallucinated outputs produced by the LLM. This reduces the effectiveness of using LLM agents to model market participants, due to the possible unpredictable random errors. In addition, by using an LLM agent to simulate investor decisions across *multiple* time-steps, these error terms can also accumulate, which could further limit the performance of the overall simulation. This has been previously observed in financial agent-based works (Zhang et al., 2024a), where the simulation results were shown to diverge greatly across multiple runs, over a trading period of only 10 days.

To deal with this limitation, we take inspiration from Monte-Carlo simulations, which are commonly used in finance (Metropolis & Ulam, 1949). We prompt the LLM over a range of prices in the input, *i.e.,* $X_1, X_2, \cdots, X_M$. For each price, we get a response $l_d(X_j)$, which can contain stochastic noise due to knowledge gaps and hallucinations. Across the price range, we get the aggregate function:

$$L_D(X) = \bar{l}_d(X) + \frac{1}{M} \sum_{j=1}^{M} \bar{\epsilon}(X_j), \tag{8}$$

where the numerical understanding $\bar{l}_d(X)$ is assumed to be fixed across runs given a frozen LLM.

Similar to how individual irrational behavior is smoothed out in a large population, we can also apply the law of large numbers in this case. As the number of price points $M$ increases, the stochastic noise from the LLM's knowledge gaps and hallucinations would tend towards a fixed mean $c$, which gives us the general internal knowledge of the LLM, plus a *constant* error offset. Formally, we then have:

$$\lim_{M \to \infty} \frac{1}{M} \sum_{j=1}^{M} \bar{\epsilon}(X_j) = c. \tag{9}$$

Therefore, the aggregated LLM function over a large range of prices would be represented by:

$$L_D(X) \approx \bar{l}_d(X) + c, \tag{10}$$

which removes the stochastic component. For an ideal market simulation, the function $L_D(X)$ should resemble a market demand curve (Schultz, 1924) when the outputs are plotted on a graph.

### 3.4 SIMULATION-BASED FINANCIAL FORECASTING

While the previous steps show that it is possible to simulate the market demand response across multiple price points, they are still not sufficient to obtain the actual price forecasts of the asset.

In economics, the price value of an entity is typically uncovered through the interaction between its demand and supply (Mankiw & Taylor, 2020). To simulate this, we repeat the above steps using a different prompt to simulate the sellers for the asset in the market $l_s(X)$. Similarly, aggregating across a range of prices, we can also obtain the agent-simulated market supply response $L_S(X)$.

Following economic theory, to obtain the price forecast for that day, we can then find the equilibrium point $y_{eq}$ in which the two curves intersect, *i.e.,* $L_D(X) = L_S(X)$. In addition, note that each demand and supply LLM responses also contain a constant error offset $c$ (shown in Equation 10), which could possibly lead to imprecisions in the price forecasts. Following this, we additionally introduce a learnt linear function $f$ to remove this. Our price forecast $\hat{y}$ can then be obtained through:

$$\hat{y} = f(y_{eq}). \tag{11}$$

Through MMARP, this step then allows us to obtain price forecasts using LLM multi-agent simulations, which differs from both traditional deep-learning methods (which only capture historical patterns) or single LLM-based reasoning methods (which could be affected by stochastic noise terms).

## 4 EXPERIMENTS

We perform two categories of experiments on MMARP. In the first, we probe the LLM agents across different price values, contextual information and agent roles to study how closely their responses match real-life demand and supply curves. Given that it is difficult to collect the real-life behavioral curves of an actual market (which would requires all transactional data points), these experiments are evaluated *qualitatively*, which is similar to many agent-based simulation works in Finance (Gao et al., 2024; Zhang et al., 2024a). In the second, we use these curves to produce market price forecasts, which can then be evaluated *quantitatively*. This was not commonly done in previous agent-related works (Zhang et al., 2024b; Yu et al., 2024a;b), which focused on profitability metrics.

Through these experiments, our work aims to answer two research questions:

1. Does the simulated response behavior of MMARP accurately replicate those in real-life markets?
2. How does MMARP perform against deep-learning and LLM methods in financial forecasting?

**Datasets** We evaluate MMARP over three datasets related to financial markets. The first dataset is stock price data for 5 large-capital stocks from the U.S. stock market, namely AAPL, MSFT, TSLA, WMT and XOM. The second dataset is exchange rate data for USD against 3 popular currencies, namely EUR, JPY and GBP. The third dataset is commodity price data for 3 items across different markets, namely Gasoline (Energy), Wheat (Agriculture) and Gold (Precious Metals). For the contextual information, we provide economic and financial news collected from Reuters[2]. In addition, tweets that mentioned each stock (Koa et al., 2024) were also provided for the stock dataset. The experiments were conducted over the period of year 2020-2022. We provide all new information between the previous and current market close times, to forecast the percentage change between the close prices based on this context. More information on the dataset can be found in Appendix A.

**Baselines** For the forecasting experiments, we evaluate the performance of MMARP against baselines from traditional deep-learning models and both general and financial-based LLMs. For deep-learning models, we explore the use of Long Short-Term Memory (LSTM) networks (Li et al., 2019), Attentive Gated Recurrent Units (GRU) (Sawhney et al., 2020) and Transformer-based models (Yang et al., 2022). For LLMs, we explore the use of GPT-4o and Mistral-v0.3 for the general models, and InvestLM (Yang et al., 2023b) and FinGPT (Yang et al., 2023a) for the models that are fine-tuned for financial tasks. Descriptions of their implementation can be found in Appendix B.

**Experimental Settings** MMARP requires an embedding-visible LLM to generate its simulated responses. We report the performance of the model using `Mistral-7B-Instruct-v0.3` in the main results, and explore other open-weights LLMs in the ablation study. On the forecasting task, the deep-learning models and linear transformation function in MMARP require fitting, which is done using a 6:1:3 data split. All forecasting results are compared over the test set. For evaluation, we compare the Mean-Squared Error (*MSE*), and the information coefficient (*IC*) between the prediction and ground truth series, a metric often used in finance prediction works (Lin et al., 2021).

---

[2]https://www.reuters.com/

## 5 RESULTS

Next, we will discuss the performance of MMARP in tackling each of the proposed research questions, evaluating the quality of its simulated data points and the accuracy of its price forecasts.

### 5.1 QUALITY OF SIMULATION

To study the accuracy of the MMARP simulated results, we study its response behavior across different numerical prices and contexts. For the plots below, each data point represents the probability of a "buy" action for the agent at each price point, given the same contextual information. This action is calculated from the LLM-generated next-token probabilities. Note that the independent and dependent axes in our plots are also swapped to emulate the supply and demand curves in economics.

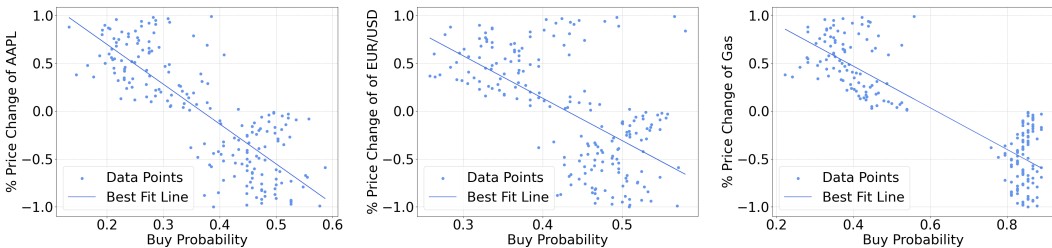

Figure 2: Examples of the generated response curves across three different domains (*i.e.,* stocks, currencies and commodities). In each plot, for each point, the prompts are the same except for the input prices. We observe the probability of Buy action from the LLM agent at different price points.

From Figure 2, we can make the following observations:

- It can be observed that the LLM's responses to the price points are not precise, which highlights the above-mentioned limitations in understanding numerical price values. For example, at some points, it is more likely for the LLM to buy the entity at a higher price, despite being provided with the same information. This observation might also indicate their limitations in simulating rational investors' actions in the market, restricting their applications in agent-based financial models.

- On the other hand, by plotting the best-fit line across all the points, we can observe a downward sloping trend, which is the same as the real-life demand curve in economics. In other words, the higher the price of the entity, it should be less likely for the LLM to choose to buy it (assuming all other factors stay the same). This line represents our aggregate function $L_D(X)$ in Equation 10, which represents the LLM's internal understanding of the market demand based on its train data.

- Across the different domains, there are differences in the agent response behaviors. For example, when gasoline prices fall, we observe that the buy probability remains relatively static at a high value. The demand for gasoline is typically inelastic (Havranek et al., 2012; Nicol, 2003), given that it is a versatile source of energy for household and transportation, and the LLM responses are likely reflecting this. The impact of non-price factors on the LLM responses will be studied next.

Next, we observe the LLM response behaviors given different contextual information in the prompts. In economics (Mankiw & Taylor, 2020), the positioning and slope of the demand curve are affected by the non-price determinants and price elasticity of demand respectively. A decrease in demand causes the curve to shift to the left, while lower price elasticity results in a less steep slope, reflecting reduced sensitivity to price changes. To observe how closely LLM response behaviors follow real-life market behavior, we *qualitatively* compare multiple plots given these different characteristics.

From Figure 3, we can make the following observations:

- The left figure compares two curves of AAPL stock across different contexts. The "Low Demand" curve shows the LLM response behavior on 30 Apr 2021, where it is discussed that there was potential *market overvaluation* of the stock. On the other hand, the "High Demand" curve shows the behavior on 8 Jul 2020, where Deutsche Bank has just *raised its price target* for AAPL stock. On the figure, we see that for the period of low demand, there is a lower buy probability for the

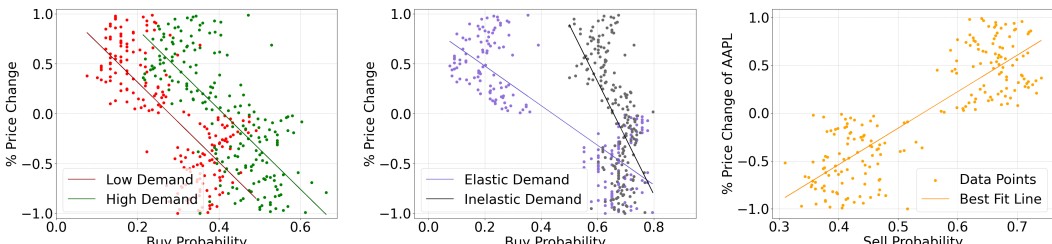

Figure 3: Examples of the generated response curves for stocks under different scenarios. On the left, the agents are prompted for buying AAPL stock on days with high and low demand. In the middle, the agents are prompted for buying stocks with different price elasticities. On the right, the agent is now in a Seller role, and we observe the probability of Sell action at different price points.

same price points, resulting in the curve positioned towards the left. Similarly, for the period of high demand, there are a higher buy probabilities, positioning the curve towards the right instead.

- The middle figure compares the curves of AAPL and WMT stocks on 27 Jul 2022. On this date, it was reported that there is a fall in demand for consumer electronics such as smartphones and tablets, which impacted the sales of both Apple and Walmart. The price elasticity for both companies are not the same, affecting the responses differently. Apple products have more substitutes and are positioned towards the luxury end, making the company more sensitive to changes in economic conditions (*i.e.,* elastic demand). Walmart, while carrying Apple products, also sell necessities such as groceries, making it less sensitive to economic changes (*i.e.,* inelastic demand).

- Finally, in the right figure, we look at the LLM agent in a different role. Here, the agent was now prompted for the probability of a "sell" action at each price point. We can see that by plotting the best-fit line, we now observe an upward sloping trend, similar to a market supply curve. Here, the higher the price of the entity, the more likely for the LLM to choose to sell. This line represents the agent-simulated market supply response $L_S(X)$, which also closely follows real-life behavior.

## 5.2 FORECASTING PERFORMANCE COMPARISON

To obtain market forecasts using MMARP simulation, we extract the intersection points between the agent-simulated demand and supply responses (see Figure 4), given the same information contexts for each day.

From Table 1, we observe that MMARP outperforms all LLMs in all metrics. Following the motivation of our work, LLMs are usually prone to hallucinations and have known gaps in their numerical knowledge, which limit their performance in this regression-based stock prediction task. Additionally, the financial LLMs are typically used for binary or percentile-based classification tasks, likely due to limitations on exact numerical reasoning.

On the other hand, MMARP did not outperform some deep-learning methods in the MSE metric for price prediction. It is possible that the poorer MSE performance is

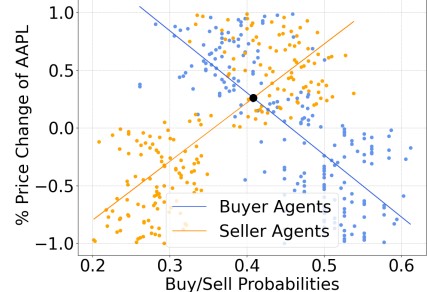

Figure 4: The interaction between buyers and seller agents (i.e., demand and supply) let us obtain the price forecasts.

due to the difference in value scales, despite the linear transformation that was done. In our experiments, the deep-learning based models were only trained on stock price values for this task, while the LLMs were originally trained on a variety of numerical values from its large data-set, with no additional fine-tuning. For example, the LLM agents sometimes predict large changes in price such as 5,000% for stock prices, which is highly unlikely for the selected stock companies in this work.

Finally, MMARP outperformed all models, including deep-learning models, on the IC metric. The IC metric shows that it can understand the price trends in a similar manner as the ground truth values (*i.e.,* the larger the price change, the larger the predicted value in general). We note that this result could be related to the earlier observation, where we show that LLM might not be precise in numerical reasoning but is able to capture general trends in the data. It might be possible to obtain better MSE through more complex transformation functions, which can be explored in future works.

Table 1: Performance comparisons of MMARP against baselines. ($\downarrow$) signifies lower is better, while ($\uparrow$) signifies higher is better. The second-best results are underlined; the best results are boldfaced.

| | Stocks | | Exchange Rate | | Commodities | |
|---|---|---|---|---|---|---|
| | MSE ($\downarrow$) | IC ($\uparrow$) | MSE ($\downarrow$) | IC ($\uparrow$) | MSE ($\downarrow$) | IC ($\uparrow$) |
| **Deep-Learning** | | | | | | |
| LSTM | $1.62\times10^{-3}$ | -0.0320 | $6.84\times10^{-5}$ | -0.007 | $9.99\times10^{-4}$ | -0.0382 |
| GRU + Att | $7.37\times10^{-4}$ | 0.0283 | $7.82\times10^{-5}$ | 0.0216 | $\mathbf{8.06\times10^{-4}}$ | 0.0033 |
| Transformer | $\mathbf{7.09\times10^{-4}}$ | -0.0536 | $6.97\times10^{-5}$ | 0.0119 | $1.01\times10^{-3}$ | -0.0112 |
| **General LLMs** | | | | | | |
| GPT-4o | $1.00\times10^{-3}$ | 0.0171 | $7.66\times10^{-5}$ | -0.0114 | $1.07\times10^{-3}$ | 0.0917 |
| Mistral-v0.3 | $1.70\times10^{-3}$ | -0.0428 | $6.70\times10^{-5}$ | 0.0135 | $1.17\times10^{-3}$ | 0.0928 |
| **Financial LLMs** | | | | | | |
| InvestLM | $6.38\times10^{-3}$ | 0.0002 | $1.86\times10^{-2}$ | 0.0002 | $1.95\times10^{-2}$ | 0.0485 |
| FinGPT | $7.44\times10^{-4}$ | -0.0270 | $7.33\times10^{-5}$ | -0.0271 | $1.03\times10^{-3}$ | 0.0213 |
| FinMA | $3.91\times10^{-3}$ | 0.0111 | $1.21\times10^{-4}$ | -0.0441 | $3.60\times10^{-3}$ | -0.0099 |
| **Ours** | | | | | | |
| MMARP | $7.12\times10^{-4}$ | **0.0630** | $\mathbf{6.67\times10^{-5}}$ | **0.0425** | $8.86\times10^{-4}$ | **0.1005** |

## 5.3 ABLATION STUDIES

To investigate the effectiveness of the model design, we additionally explored different methodologies for the components of MMARP, in order to study their impact on the overall model behavior. Three components were studied: the prompt design, the type of LLM used and the simulation scale.

**Prompt Design** To obtain the probability for Buy or Sell, we provide the LLM agents with binary options for their responses. For the buyer agent, it is asked whether the asking price is *Too Expensive* or *Too Cheap*, where the second option gives the Buy probability. The seller agent is asked whether the bid price is *Too Low* or *Too Good*, where the second option would give us the Sell probability.

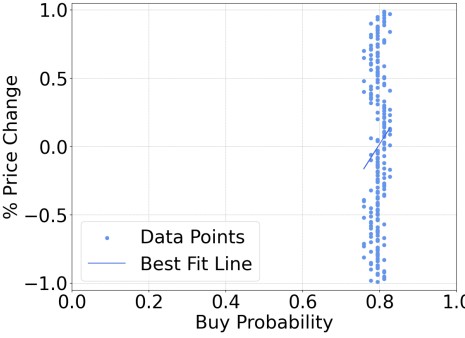
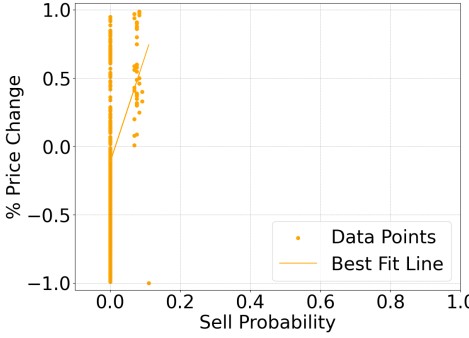

Figure 5: Given the choice between *Too Expensive* and *Just Right*, the LLM would tend towards choosing the more neutral *Just Right* option, resulting in mostly simulated buyers.

Figure 6: Given the choice between *Too Low* and *Too High*, the LLM unlikely choose *Too High*, given that it does not make logical sense. This results in few simulated sellers.

The set of binary options were purposefully designed to be extreme opposites in order to "force" the LLM to pick a side, such that the demand plot can be observed. By allowing the LLM to choose between more reasonable options such as whether the price is *Too Expensive* or *Just Right* (where those who selected *Just Right* would be simulated as buyers), LLMs tend to prefer the more neutral *Just Right* option, resulting in a high Buy probability regardless of given price point (see Figure 5).

Additionally, the semantic meaning of the options also have to be taken into account. For example, for the seller agent, given the opposite choice between whether the price is *Too Low* or *Too High* (where those who did not select *Too Low* would be simulated as sellers), close to zero LLM agents would pick *Too High*, as no offered price would logically to be "too high" to any potential seller. This would result in close to zero Sell probability regardless of any given price point (see Figure 6).

**Different Large Language Models** MMARP requires an embedding-visible LLM to produce its simulated responses, by utilizing the generated next-token probabilities. We also explore the use of other open-weights LLMs to implement MMARP, which include *Qwen2-7B-Instruct*, *Gemma-2-9b-it* and *Meta-Llama-3.1-8B-Instruct*. These models were chosen to represent the different methods of tokenizing numbers, and the model parameters were chosen to be as similar as possible. Table 2 reports

Table 2: Ablation study of MMARP utilizing different open-weights LLMs as its base.

|  | MSE ($\downarrow$) | IC ($\uparrow$) |
|---|---|---|
| MMARP-Qwen | $7.34\times10^{-4}$ | 0.0301 |
| MMARP-Llama | $\mathbf{7.04\times10^{-4}}$ | 0.0407 |
| MMARP-Gemma | $7.17\times10^{-4}$ | 0.0506 |
| MMARP-Mistral | $7.12\times10^{-4}$ | **0.0630** |

the ablation results on the stock dataset. We can observe that the MSE of all models lie close to each other, which highlights the consistency of the MMARP method. Any sources of stochastic errors are mostly removed from the repeated prompting, and the final predictions are representative of the LLM understanding of financial values learnt from their original corpora of training data. This result differs from previous works on financial agent-based simulation, which showed that different LLMs can produce vastly different results (Zhang et al., 2024a). On the IC metric, the two best performing models are Gemma and Mistral, which use the same numerical embedding method (*i.e.,* every digit is its own token). The Mistral LLM family is also often discussed to be good at mathematical tasks (McNichols et al., 2024) and used to tune mathematical-based LLMs (Mitra et al., 2024; Tang et al., 2024). This could point to the effectiveness of this digit tokenizing method and its usefulness in other numerical-based tasks such as understanding financial values, which is observed in MMARP.

**Scale of Simulation** In MMARP, by prompting the LLM agent across a range of price values, we minimize the stochastic error terms caused by the LLM's knowledge gaps and hallucinations. This result stems from the law of large numbers: as number of trials in a probability-based experiment increases, the sample average of the outcomes will converge to the expected value (in our case, a constant error term). To study the effectiveness of repeated sampling in MMARP, we plot the number of prompted price values against the achieved information coefficient (IC) of the price forecasts. This ablation study is conducted over the stock price dataset.

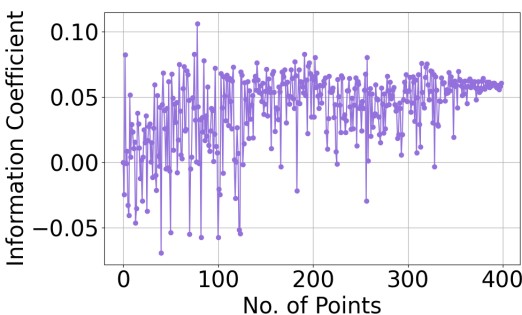

Figure 7: Information coefficient of forecasts from different number of prompted price values.

From Figure 7, when the number of simulated price points is small, we can observe high variance in the IC results. On average, the IC starts low but increases with the number of points, occasionally achieving high values likely due to random chance. As the number of prompted price points grows bigger, the IC then stabilizes towards a fixed value, converging at the maximum possible value obtainable given the knowledge base of the LLM.

## 6 CONCLUSION

In this work, we explored the use of LLMs to model real-life market participants to produce simulated price market forecasts. For this task, we highlighted two challenges: the stochastic actions caused by non-rational individuals in the market, and the stochastic errors caused by the numerical knowledge gaps and possible hallucinated outputs of LLMs. To handle these challenges, we propose a prompt design framework, MMARP, which simulate repetitive prompting using LLM-generated next-token weights and probe the LLM across a range of price inputs to obtain an aggregated response function that can represent actual market behavior. To verify the effectiveness of MMARP, we conducted extensive experiments across three market-based datasets on stock prices, exchange rates and commodity prices. Empirically, we can observe that the LLM's responses to individual price points are not precise, which would limit their effectiveness if they are used directly in agent-based financial models. However, we also find that its aggregate response function contains the traits of actual market behavior, which could be exploited to produce realistic market simulations. To evaluate this quantitatively, we then produce price forecasts using the intersection point of the response functions of LLM agents in buyer and seller roles. We find that our simulated price forecasts show strong competitive results, and can also outperform other baseline models using the IC metric.

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

# A  DATASET

To verify the effectiveness of MMARP, we conducted extensive experiments across three market-based datasets on stock prices, exchange rates and commodity prices. The data for stock prices and exchange rates can be found on Yahoo Finance, while the data for commodities can be found on Kaggle. We collect data for the period of 01/01/2020 to 31/12/2022, for a total of 755 market days.

For the forecasting task, in order to compare with the trained deep-learning, we split the data into a train-valid-test ratio of 6:1:3. We then evaluate on the test set only, regardless if the model requires training. The duration and number of trading days for the split data sets can be found in Table 3.

For the contextual information for all three tasks, we provide the LLM with economic and financial news collected from Reuters. In addition, tweets that mentioned each stock were also provided for the stock dataset, given that stock prices are often also affected by public sentiment (Pagolu et al., 2016). This is taken from the *SN2* dataset (Koa et al., 2024), which follow the same format as the popular *StockNet* dataset (Xu & Cohen, 2018), updated for the year 2020-2022. We provide the new information that are received between the previous and current day's market close time, and forecast the change between these close prices. Some statistics of the text datasets can be found in Table 4.



Table 3: Price dataset statistics.

|           | Duration                | Trading Days |
|-----------|-------------------------|--------------|
| Train Set | Jan 01 '20 - Oct 20 '21 | 450          |
| Valid Set | Oct 21 '21 - Feb 07 '22 | 75           |
| Test Set  | Feb 22 '22 - Dec 31 '22 | 226          |

Table 4: Text dataset statistics.

|                   | Tweets | News   |
|-------------------|--------|--------|
| Avg. texts per day| 16     | 69     |
| Max texts per day | 1,599  | 187    |
| Total no. of texts| 17,536 | 76,217 |



# B  BASELINES

We evaluate the performance of MMARP against baselines from traditional deep-learning models and both general and financial-based LLMs. We note that there are few available baselines that directly perform numerical stock prediction, or use text data only. For most models, we either adapt the last layer in order to do regression prediction, or otherwise remove the other input modalities.

**LSTM** (Li et al., 2019)**:** The selected LSTM model is the closest model to our task, which uses text data to do regression-based prediction. This specific model also uses differential privacy techniques to hide sensitive information within the text data, which is beyond the scope of our work. We keep the time-series component in this model, allowing it more information than our MMARP method.

**GRU + Attention** (Sawhney et al., 2020)**:** This model combines GRU with an attention mechanism to do stock movement prediction. In particular, the attention layer enhances the model's ability to capture key information from noisy, unstructured text data. Similarly, we keep the time-series component for utilizing GRU. We adapt the binary classification layer to do regression predictions.

**Transformer** (Yang et al., 2022)**:** This is a transformer-based architecture tailored for financial forecasting. It excels in processing hierarchical, numeric-heavy financial data and performing multi-task learning using text and audio data. However, we remove the audio processing component due to a lack of data for this modality. Similarly, we adapt the final layer to do regression predictions.

**Llama 3.1**: A refined version of Meta's Llama 3 (Dubey et al., 2024), optimized for diverse language understanding and reasoning tasks. We directly prompt the LLM to produce a numerical prediction.

**Mistral-v0.3**: An updated version of Mistral v0.1 (Jiang et al., 2023), which have shown good results on math-related tasks. Similarly, we directly prompt the LLM to produce a numerical prediction.

**InvestLM** (Yang et al., 2023b)**:** One of the earliest LLMs for financial-forecasting. The model was trained to produce binary predictions. For our task, we prompt the LLM for a numerical prediction.

**FinGPT** (Yang et al., 2023a)**:** Another financial-trained LLM, which can do stock forecasting in percentiles. However for our task, we prompt the LLM for numerical prediction for fair comparison.

**FinMA** (Xie et al., 2023)**:** An open-sourced financial LLM trained to do general financial tasks, such as sentiment analysis, financial QA, *etc*. We prompt the LLM to make numerical predictions.

## C EXAMPLE CONTEXT

Here, we provide the contextual information used in Section 5.1 to generate the LLM responses.

Contextual Information for AAPL on 2020-07-20

```
- Apple (AAPL) has opened a megastore in Beijing amidst criticism from
    the U.S.
- AAPL is currently experiencing below-average trading volume.
- Historical performance data indicates that AAPL has had an average
    increase of 1.33% five days after similar trading conditions, with a
    standard deviation of 6.26%. The worst performance was a decrease of
    47.29%, while the best was an increase of 31.53%.
- Over a ten-day period, the average increase is 2.72% with a standard
    deviation of 9.52%, and the worst performance was a decrease of
    54.19%, while the best was an increase of 35.52%.
- Over a thirty-day period, the average increase is 6.70% with a
    standard deviation of 20.95%, with the worst performance being a
    decrease of 96.43% and the best an increase of 74.26%.
- There are high expectations for AAPL's stock performance, with some
    predictions suggesting it could reach $450 by fall.
- AAPL is favored to extend higher in the near term according to some
    analysts.
- The stock is currently experiencing significant options activity, with
    large call options being purchased.
- AAPL is considered a key player in the tech sector, with its
    performance impacting broader market indices like the SPY and QQQ.
- The company is expected to benefit from upcoming 5G phone launches and
    advancements in technology.
```

Contextual Information for AAPL on 2021-04-30

```
- Apple ($AAPL) reported a remarkable revenue growth of 54%
    year-over-year for Q1, marking the highest growth rate since 2012.
- The company's revenue for Q1 reached $89.6 billion, significantly
    surpassing expectations.
- iPhone sales were particularly strong, generating $47.9 billion, a 66%
    increase compared to the previous year.
- Apple announced a 7.3% increase in its quarterly dividend, raising it
    to $0.22 per share.
- The company is actively engaging in share repurchases, committing a
    substantial portion of its operating cash flow to this effort.
- Apple's market capitalization is approximately $2 trillion.
- Despite strong earnings, the stock experienced a decline in price
    following the announcement, indicating potential market
    overvaluation.
- Apple is facing antitrust charges from EU regulators related to its
    App Store practices, which could impact its operations in the music
    streaming market.
- The stock is currently a focus among traders, with significant options
    activity noted.
```

Contextual Information for AAPL on 2022-07-27

```
- Bank of America has cut Apple's price target from $200 to $185, citing
    concerns over foreign exchange impacts on sales.
- Apple is currently discounting iPhones in China for a limited time.
- AAPL's stock has shown mixed performance, with a recent decline of
    approximately 0.36% to 0.9% in various reports.
- Upcoming earnings reports for Apple are anticipated, with significant
    attention from investors and analysts.
- There is notable bearish sentiment in the options market for AAPL,
    with a higher percentage of put options compared to call options.
- Apple has filed patents related to self-driving and vehicle software,
    indicating ongoing innovation in the automotive sector.
```

– Analysts are closely monitoring AAPL's performance in the context of broader market trends and economic data releases.

### Contextual Information for WMT on 2022-07-27

Walmart's stock ($WMT) has recently experienced significant volatility, primarily due to a major profit warning that led to a sharp decline in its share price, dropping approximately 9.5% in after-hours trading. This decline resulted in a loss of about $36 billion in market capitalization. The company cut its full-year profit forecast, which has raised concerns among investors about the overall health of the retail sector, leading to similar declines in other retailers like Target and Costco.

Key factors contributing to Walmart's stock performance include:
– A decrease in demand for certain consumer electronics, such as smartphones and tablets.
– Rising inventories and price cuts implemented by Walmart in response to inflationary pressures.
– The Walton family's fortune decreased by $12.9 billion due to the stock's decline.
– Despite the profit warning, Walmart reported a projected increase in same-store sales of 6%, indicating some resilience in its core business.

Analysts have reacted by adjusting their price targets for Walmart, with some lowering their expectations significantly. The overall sentiment in the market suggests a cautious outlook for Walmart and the retail sector as a whole, with fears of a potential recession impacting consumer spending.

