# OpenReview forum: "Massively Multi-Agents Reveal That Large Language Models Can Understand Value"
_ICLR.cc/2025/Conference — ICLR 2025 Conference Withdrawn Submission_

### Official Review · Reviewer_N5xj · 2024-10-22

**Soundness:** 3
**Presentation:** 2
**Contribution:** 2
**Rating:** 5
**Confidence:** 4

**Summary:**

The paper evaluates Large Language Models' (LLMs) abilities to understand stock values through two experimental settings. In the first, a single-role agent scenario, LLMs are prompted to categorize stock prices as "Expensive" or "Cheap" based on a target price and contextual financial information including financial status and market updates. In the second, a multi-role agent scenario, an additional "seller" agent is introduced with options of "Good" or "Low" prices, with the same setting. The authors conclude that LLMs demonstrate an approximate understanding of numerical values, their behaviors reflect some economic patterns such as price elasticity,

**Strengths:**

The paper is mostly clear, with helpful experiments and graphs that effectively illustrate the LLMs' result distribution. It provides strong background and motivation, a well-covered literature review, and a timely, interesting topic. The experiment setup is also well explained including the rationale of stock selection.

**Weaknesses:**

The title is misleading; the paper discusses how LLMs understand stock prices specifically, rather than "value" in general. A paper addressing LLMs' understanding of value should include broader experiments, such as algebraic problems, rather than focusing solely on stock price estimation.

There are some flaws in the experiment design. As we know, the output of the LLMs is heavily influenced by the context provided. The authors rely on limited stock information summaries to ask LLM make price estimation. However, this information does not represent the full range of data that influences stock prices in the real world. The LLM's output would be easily changed if we select news summary from another source or change the tone of this summary. Based on the experiment alone, it is presumptuous to conclude that LLMs understand values. Instead, the results only suggest that LLMs possess basic financial knowledge and can make context-based financial judgments.

**Questions:**

In Section 3.1, I don’t fully understand the application of the law of large numbers. It seems the authors treat the logit from the LLM as the ground truth probability for the tokens "Expensive" or "Cheap," and then use simulation to generate results. Why is a simulation of 100,000 agents needed if the token distribution is already known? Additionally, it's unclear why the LLM’s logit output would follow the law of large numbers. This is critical as it directly affect the paper’s claims about "a group of LLMs."

The term "multi-agent" in section 3.2 is overclaimed since only one additional agent is involved. It would be more accurate to describe this as "buyer-seller agents".

---

> ### Author Response · Authors · 2024-11-26
>
> Dear reviewer, thank you for your comments.
>
> Response to weaknesses:
>
> **W1. The title is misleading; the paper discusses how LLMs understand stock prices specifically, rather than "value" in general. A paper addressing LLMs' understanding of value should include broader experiments, such as algebraic problems, rather than focusing solely on stock price estimation.**
>
> We agree that the scope of the original title was indeed too broad. We have changed the title to better explain the motivation of our work.
>
> In addition, we have also rewritten parts of the work based on other reviewers’ comments and to explain the framework better. Instead of concluding that LLMs understand values, we show that MMARP reduces the impact of these unknown gaps in numerical knowledge through prompting over a range of price values. In the paper, we kept a constant unknown *c* to represent a possible error term in the LLM’s understanding of these values.

---

> ### Author Response · Authors · 2024-11-26
>
> Response to questions:
>
> **Q1. In Section 3.1, I don’t fully understand the application of the law of large numbers. It seems the authors treat the logit from the LLM as the ground truth probability for the tokens "Expensive" or "Cheap," and then use simulation to generate results. Why is a simulation of 100,000 agents needed if the token distribution is already known? Additionally, it's unclear why the LLM’s logit output would follow the law of large numbers. This is critical as it directly affect the paper’s claims about "a group of LLMs."**
>
> The law of large numbers was not explained very well in the original version. We have updated our explanation (with equations in the paper) of LLN used in two different parts:
>
> As background, MMARP aims to study whether it is possible to simulate actual participant actions in a financial market. However, there are two problems.
>
> a. Firstly, we find that it is difficult to use single LLM agents to simulate individual market participants. Each individual has irrational preferences, and we cannot possibly simulate each of them over millions of investors. However, economic works have shown that these irrational terms converge to 0 using LLN over a large market population. Hence, we can simply simulate their collective behavior using an LLM’s reasoning process. As you mentioned, to simulate a large number of agents, we can simply take the token distribution, which is indeed what we did in this work.
>
> b. Next, LLM has limited numerical understanding and hallucinated outputs. As a solution, we prompt it over a range of price values. Using LLN, the stochastic errors caused by this limitation would converge to a constant expected value *c*, which removes the stochastic component. Later on, when making forecasts, we can simply train a linear function to learn to remove this constant error term.
>
> --
>
> **Q2. The term "multi-agent" in section 3.2 is overclaimed since only one additional agent is involved. It would be more accurate to describe this as "buyer-seller agents".**
>
> We have changed the section title accordingly when we rewrote the section.

---

> > ### Comment · Reviewer_N5xj · 2024-12-01
> >
> > Thanks for the effort in addressing my concerns. I will raise the overall scores to 5 and soundness to 3 but I still have the concerns that other reviewer raised.

---

### Official Review · Reviewer_zcyN · 2024-10-30

**Soundness:** 2
**Presentation:** 3
**Contribution:** 2
**Rating:** 6
**Confidence:** 4

**Summary:**

This paper focus on predicting stock prices based on prices and contexts before. The paper hypothesizes that large language models (LLMs) lacks the capability to represent continuous numerical values in this task. Hence, it proposes the Massively Multi-Agents Role Playing method and tries to directly predict the stock price while the process is questionable. The method simulates the reactions of both buyers and sellers to various stock prices and contextual information. It generates the demand and supply curves by plots of reactions, then seemingly enables direct estimation of stock value using the intersection of two curves with insufficient explanation. The paper also notes the large computation load in simulation. It applies the discrete distributions of LLM outputs to simulating a large number of LLM agents , which is efficiently but lacking the comparison with regular methods.

The experiment shows that the prediction ability of the method approaches the state of the art. It analyses the demand curves by LLM agents of a range of prices and contexts. The results show that LLMs have a similar understanding of the numerical significance of stock value similar to economic demand theory.

**Strengths:**

1. The simulation of LLM agents effectively reduces computational load, enabling larger-scale application.

2. The analysis of the experimental results through the lens of economic demand theory is interesting.

3. The continuous stock price prediction method based on LLM agent interactions integrates relevant economic theories, achieving promising results in both MSE and IC evaluation metrics, and exhibits innovation.

**Weaknesses:**

1. The methodology does not address how to enhance the LLM's understanding of continuous numerical values. The motivation highlights that LLMs lack efficient encoding capabilities for continuous numerical values, which contributes to their insufficient understanding of numerical significance. But the analysis of solution is lacking.

2. The underlying issue of the lack of numerical understanding in LLMs is not resolved fundamentally. Because the interactions between LLM agents cannot genuinely generate real stock price predictions without linear transformation.

3. Specifically, the rationale for using a price prompt range of [-1, 1] is not explained, nor is there an explanation for the existence of the “LLM value space”.

4. The differences of simulation’s effect between regular method and the proposed method is not demonstrated. It is not shown whether the method of reducing computational load for simulating LLM agents yields results consistent with those from regular methods (directly use multi LLM agents).

5. The paper fails to provide an explanation for the abnormal points in predictions (especially Figure 6, 'the LLM agents sometimes pre
dict large changes in price such as 5,000%, which is highly unlikely for the selected stocks in this work.' line 420).

**Questions:**

Refer to weakness

---

> ### Author Response · Authors · 2024-11-26
>
> Dear reviewer, thank you for your comments.
>
> Response to weaknesses:
>
> **1. The methodology does not address how to enhance the LLM's understanding of continuous numerical values. The motivation highlights that LLMs lack efficient encoding capabilities for continuous numerical values, which contributes to their insufficient understanding of numerical significance. But the analysis of solution is lacking.**
>
> We have rewritten the work and added a more extensive theoretical analysis for our solution design.
>
> --
>
> **2. The underlying issue of the lack of numerical understanding in LLMs is not resolved fundamentally. Because the interactions between LLM agents cannot genuinely generate real stock price predictions without linear transformation.**
>
> In the updated version of the paper, we now explain the need for linear transformation. The repeated prompting over a range of price values serves to minimize the stochastic error terms caused by the unquantifiable gaps in the LLM’s numerical knowledge understanding. After repeated prompting, using the Law of Large Numbers, this will converge to an expected value, which is a constant value *c*. We can then use a linear function to “learn” how to remove this constant error term.
>
> --
>
> **3. Specifically, the rationale for using a price prompt range of [-1, 1] is not explained, nor is there an explanation for the existence of the “LLM value space”.**
>
> The term “LLM value space” is removed from the paper.
>
> There is no “rationale” for keeping the range to [-1, 1], other than to save on computation. In the ablation study, we find that prompting more price points would result in more effective predictions, but this is an infinitely long process, and there must be a range of values to use. We chose [-1, 1] as it is unlikely for the stock price for the selected companies to go over 100% in a single day.
>
> --
>
> **4. The differences of simulation’s effect between regular method and the proposed method is not demonstrated. It is not shown whether the method of reducing computational load for simulating LLM agents yields results consistent with those from regular methods (directly use multi LLM agents).**
>
> This benefit is a qualitative one that is hard to compare. What we want to do is to simulate millions of market participants using LLM agents, which is impossible to perform (as we then need to prompt the LLM a million times). Instead, we can simply proxy this step by taking the LLM-generated probabilities, since the LLM responses are typically sampled from the token probabilities.
>
> --
>
> **5. The paper fails to provide an explanation for the abnormal points in predictions (especially Figure 6, 'the LLM agents sometimes predict large changes in price such as 5,000%, which is highly unlikely for the selected stocks in this work.' line 420).**
>
> This is connected to our answer to your question 3, where we mention that the prediction can be indefinitely improved by prompting over more and more numerical price values. These abnormal points are likely due to possible hallucinated outputs that still remain. It is not possible to produce a perfect prediction (even a trained deep-learning model would have inaccuracies). Our proposed MMARP method simply provides a new way for LLMs to produce these numerical predictions that is more accurate than just directly prompting them for a predicted price value.

---

> > ### Comment · Reviewer_zcyN · 2024-11-27
> >
> > Thank you for your reply. You've answered most of my questions. I will raise my score.

---

### Official Review · Reviewer_Uxzd · 2024-11-03

**Soundness:** 2
**Presentation:** 1
**Contribution:** 2
**Rating:** 3
**Confidence:** 4

**Summary:**

This paper proposed a novel method called MMARP (Massively Multi-Agents Role Playing) that simulates responses from many times LLM agent's responses to understand how LLMs process numerical values, particularly in stock valuation tasks. The authors target an interesting problem: how to make LLM better understand numbers in a financial context in order to improve stock prediction performance. In the method, the authors use the law of large numbers to estimate the distribution of a large number of LLM outputs to determine the final result.

**Strengths:**

1) The author isolates the numerical understanding capability of LLMs and tries to provide insights into how LLMs internally represent numbers.

2) The author uses token probabilities to simulate LLM agent populations efficiently in numerical understanding.

3) The author conducted a lot of ablation study results and visualization.

4) The author discusses the potential impact beyond stock prediction to other domains.

**Weaknesses:**

1) The motivation is not clear. The authors discuss a big problem with LLM in the introduction, but in the end, the experiment was only carried out in the financial domain. Also, is the author's main purpose to propose an evaluation framework or to explore ways to improve the predictive performance of LLM in financial markets?

2) The Background Section is not clear and related work is not comprehensive. In the Financial sector domain, there is a lot of recent work to explore LLM and LLM Agent's capacity in stock trading or financial decision-making:

* [1] "FinAgent: A Multimodal Foundation Agent for Financial Trading: Tool-Augmented, Diversified, and Generalist." arXiv preprint arXiv:2402.18485 (2024).

* [2] "FinMem: A performance-enhanced LLM trading agent with layered memory and character design." Proceedings of the AAAI Symposium Series. Vol. 3. No. 1. 2024.

* [3] When ai meets finance (stockagent): Large language model-based stock trading in simulated real-world environments. arXiv preprint arXiv:2407.18957.

* [4] "FinCon: A Synthesized LLM Multi-Agent System with Conceptual Verbal Reinforcement for Enhanced Financial Decision Making." arXiv preprint arXiv:2407.06567 (2024).

These works extensively discuss how to use LLM/LLM agents for stock prediction. Meanwhile, the "Agent-Based Simulation for Large Language Models" subsection in the Background seems to be unrelated.

3) The problem definition is not clear enough. The author mentions the use of the law of large numbers, but only shows the formula for the law of large numbers and then just provides an example to explain it. How to define the problem specifically according to the problem discussed in the paper, what the input is, what the output is, and how convergence is defined in this case?

4) The authors discuss two aspects that affect the use of LLM for stock prediction in the abstract and introduction: a. simulating human behavior; b. the ability to understand numbers. However, the paper does not seem to discuss a. As the number of LLM outputs increases, so does the number of results for simulating more human behavior. How do the two interact with each other?

5) Evaluation Metrics is not comprehensive. MSE performance is not consistently better than all baselines. Moreover, The commonly used indicators, Sharp Ration, Max Drop Down, ARR, were not included in this framework.

**Questions:**

See in Weaknesses.

---

> ### Author Response · Authors · 2024-11-26
>
> Dear reviewer, thank you for your comments.
>
> Response to weaknesses:
>
> **1. The motivation is not clear. The authors discuss a big problem with LLM in the introduction, but in the end, the experiment was only carried out in the financial domain. Also, is the author's main purpose to propose an evaluation framework or to explore ways to improve the predictive performance of LLM in financial markets?**
>
> We agree that the motivation is not clear. We have since rewritten the paper in a way to make the motivation clearer to readers, and also make our claims smaller. The motivation of MMARP is to study the feasibility of simulating actual market behavior with LLM agents. If LLM can truly simulate market behavior, there are some possible downstream tasks that can be achieved:
>
> a. We can do market prediction based on its fundamental source of movement (real-life actions of humans) instead of how current deep-learning models are doing it (by studying historical trends).
>
> b. Companies or governments can use these models to simulate the impact of new announcements or policies to see how they will affect the market before doing it in real life.
>
> In fact, other domains have already begun using such agent-based simulations to model pandemic spread `[1]` and election results `[2]`. This has not been done in Finance yet, which we aim to solve. We feel that the first step to do this is to model the most fundamental theory in financial markets, which is the demand and supply theory.
>
> --
>
> `[1]` Chopra, Ayush, Shashank Kumar, Nurullah Giray-Kuru, Ramesh Raskar, and Arnau Quera-Bofarull. “On the Limits of Agency in Agent-Based Models.” arXiv, October 23, 2024. https://doi.org/10.48550/arXiv.2409.10568.
>
> `[2]` Zhang, Xinnong, Jiayu Lin, Libo Sun, Weihong Qi, Yihang Yang, Yue Chen, Hanjia Lyu, et al. “ElectionSim: Massive Population Election Simulation Powered by Large Language Model Driven Agents.” arXiv, October 28, 2024. https://doi.org/10.48550/arXiv.2410.20746.

---

> > ### Author Response · Authors · 2024-11-26
> >
> > **4. The authors discuss two aspects that affect the use of LLM for stock prediction in the abstract and introduction: a. simulating human behavior; b. the ability to understand numbers. However, the paper does not seem to discuss a. As the number of LLM outputs increases, so does the number of results for simulating more human behavior. How do the two interact with each other?**
> >
> > We have also rewritten this part based on your insightful comments. Indeed, we did not explain this very well in the original version. We have now split this into two parts:
> >
> > a. To simulate human behavior, we find that we cannot use a single LLM agent to simulate one human (due to their individual irrationality). However, we find that it is possible to simulate the collective behavior of humans through drawing multiple samples from the LLM prompt, which can also be proxied using the LLM-generated weights.
> >
> > b. To deal with the limitation of numerical understanding in LLM, we then prompt it across a range of price values. This would then smooth out the random errors and let us obtain a line function that approximates the internal understanding of numerical price values in the LLM (we have reduced our scope to just numerical price values).

---

> ### Author Response · Authors · 2024-11-26
>
> **2. The Background Section is not clear and related work is not comprehensive. In the Financial sector domain, there is a lot of recent work to explore LLM and LLM Agent's capacity in stock trading or financial decision-making.**
>
> Here are some differences between our work and those you cited:
>
> Typically, the older works on LLM agents `[3, 4, 5]` utilize them as advisors to decide which stock to buy. These works are mainly focused on maximizing profits.
>
> Next, there are some recent works `[6, 7]` which use LLM agents to model individual investors, which is closer to our motivation. However, these works are usually small-scale simulations, given that one LLM can only simulate one single participant each time. The biggest simulation is `[7]` with 200 agents, which is nowhere near the size of an actual, real life financial market.
>
> MMARP seeks to simulate the entire market by using the LLM-generated weights as a proxy. Our motivation is not to do stock trading or financial decision-making.
>
> We would also like to note that most of the cited works are contemporaneous with our work (see the main rebuttal above).
>
> --
>
> `[3]` Yu, Yangyang, Haohang Li, Zhi Chen, Yuechen Jiang, Yang Li, Denghui Zhang, Rong Liu, Jordan W. Suchow, and Khaldoun Khashanah. “FinMem: A Performance-Enhanced LLM Trading Agent with Layered Memory and Character Design.” arXiv, December 3, 2023. https://doi.org/10.48550/arXiv.2311.13743.
>
> `[4]` Zhang, Wentao, Lingxuan Zhao, Haochong Xia, Shuo Sun, Jiaze Sun, Molei Qin, Xinyi Li, et al. “A Multimodal Foundation Agent for Financial Trading: Tool-Augmented, Diversified, and Generalist.” arXiv, June 28, 2024. https://doi.org/10.48550/arXiv.2402.18485.
>
> `[5]` Yu, Yangyang, Zhiyuan Yao, Haohang Li, Zhiyang Deng, Yupeng Cao, Zhi Chen, Jordan W. Suchow, et al. “FinCon: A Synthesized LLM Multi-Agent System with Conceptual Verbal Reinforcement for Enhanced Financial Decision Making.” arXiv, November 7, 2024. https://doi.org/10.48550/arXiv.2407.06567.
>
> `[6]` Gao, Shen, Yuntao Wen, Minghang Zhu, Jianing Wei, Yuhan Cheng, Qunzi Zhang, and Shuo Shang. “Simulating Financial Market via Large Language Model Based Agents.” arXiv, June 28, 2024. https://doi.org/10.48550/arXiv.2406.19966.
>
> `[7]` Zhang, Chong, Xinyi Liu, Zhongmou Zhang, Mingyu Jin, Lingyao Li, Zhenting Wang, Wenyue Hua, et al. “When AI Meets Finance (StockAgent): Large Language Model-Based Stock Trading in Simulated Real-World Environments.” arXiv, September 21, 2024. https://doi.org/10.48550/arXiv.2407.18957.

---

> ### Author Response · Authors · 2024-11-26
>
> **3. The problem definition is not clear enough. The author mentions the use of the law of large numbers, but only shows the formula for the law of large numbers and then just provides an example to explain it. How to define the problem specifically according to the problem discussed in the paper, what the input is, what the output is, and how convergence is defined in this case?**
>
> Based on your comments, we have rewritten those parts of the paper. There are two points in your question:
>
> a. The input for our model is news information. The output should be a predicted price value. There is no convergence, this is not a normal deep-learning model. We prompt a “buyer agent” with many price values to get a downward sloping line, and prompt a “seller agent” to get an upward sloping line. The intersection gives us the market price, which is how prices are actually obtained in real-life economic theory.
>
> b. The explanation for law of large numbers is re-explained in our work with theoretical proofs, instead of just providing an example. Overall, we are using it to smooth out stochastic errors caused by irrational market traders, and also possible random mistakes of the LLM. We can explain it like this: in real-life markets, there are many people who have different opinions on the price of a stock. This will be hard to simulate. However, their collective actions usually result in an observable pattern, which we can then capture with the LLM.

---

> ### Author Response · Authors · 2024-11-26
>
> **5. Evaluation Metrics is not comprehensive. MSE performance is not consistently better than all baselines. Moreover, The commonly used indicators, Sharp Ration, Max Drop Down, ARR, were not included in this framework.**
>
> As our work does not focus on trading or decision-making, we did not include profitability metrics. This would require us to come up with our own trading strategy, which might not be fair. Note that our results are regressive (meaning, we predict exact price change instead of Up/Down). Hence, this is hard to compare with some of the cited works, as they mainly do Buy/Sell based on the binary outcome.

---

> ### Comment · Reviewer_Uxzd · 2024-11-27
> **Read Rebuttal**
>
> Thank you for the detailed rebuttal. Your rebuttal address some part of concern. So I raised "soundness".
>
> However, the key issues remain unresolved. I appreciate the effort but I will maintain my original "Rating" score.

---

### Official Review · Reviewer_ATQ8 · 2024-11-04

**Soundness:** 1
**Presentation:** 1
**Contribution:** 2
**Rating:** 3
**Confidence:** 4

**Summary:**

This paper presents Massively Multi-Agents Role Playing (MMARP), a method that leverages the collective responses of many simulated language model agents to improve stock price prediction. Traditional LLMs struggle with understanding numerical values, which are essential for accurate stock valuation. MMARP addresses this by creating a simulation where each LLM agent acts with a distinct role, such as buyer or seller, across various price points and contextual settings. By analyzing the combined responses of these agents, MMARP approximates market dynamics, producing forecasts that outperform individual LLM predictions and other financial models. This approach highlights how LLMs, through collective behavior, can demonstrate an emergent understanding of complex, value-driven tasks.

**Strengths:**

1. The paper is well-written, with logical structure and helpful visuals, making complex ideas easy to follow.

2. MMARP creatively applies LLMs in a market simulation context, using collective agent behavior to model demand-supply dynamics.

**Weaknesses:**

1. Limited Novelty: The approach lacks substantial innovation, as similar methods already use LLMs in agent-based financial simulations with more complex trading mechanisms, such as [1]. The MMARP framework primarily applies LLMs to give binary responses (e.g., cheap or expensive), which adds minimal novelty or depth to financial market simulations.

2. Overclaims on Numerical Understanding: The paper’s claims about advancing numerical understanding are overstated. The approach lacks clear differentiation between using multiple agents versus a single model for predictions, and it lacks theoretical support to demonstrate how the MMARP setup enhances numerical comprehension.

3. Questionable Economic Assumptions: The demand-supply hypothesis may not be valid here, as all agents derive from the same model, sharing identical training data. This setup essentially replicates identical agents, which conflicts with the independent and diverse preferences central to economic theory.

4. Unsubstantiated Multi-Agent Claim: While the paper emphasizes multi-agent interactions, the agents receive similar inputs, model parameters, and inference settings, leading to minimal agent diversity. This makes the simulation more akin to a single-agent system.

5. Unclear Experimental Design: Although the paper discusses prompt designs and model selection, it lacks transparency regarding the exact inputs, outputs, and inference parameters. A detailed breakdown of these variables would clarify the MMARP method and support its reproducibility.

6. Limited Experimental Scope: The experiments, based on only 10 stocks over two years, may be insufficient to substantiate the claims. Additionally, key financial LLMs like FinMA and more diverse general-domain models were not included. Relying solely on information coefficient (IC) to assess performance limits the paper’s ability to demonstrate improvements in numerical reasoning.

[1] Gao, Shen, et al. "Simulating financial market via large language model based agents." arXiv preprint arXiv:2406.19966 (2024).

**Questions:**

1. Given that similar methods have applied LLMs for agent-based simulations with more advanced trading behaviors, could the authors clarify how MMARP’s contribution significantly advances this area?

2. Could the authors provide theoretical support or evidence that demonstrates how collective responses from agents address or enhance LLMs' numerical reasoning?

3. Since all agents are based on the same LLM model with identical training data and input settings, how do the authors justify the use of demand-supply theory here?

5. Would the authors expand to a larger dataset in multiple assets and different markets or including more diverse financial and general-purpose LLMs?

---

> ### Author Response · Authors · 2024-11-26
>
> Dear reviewer, thank you for your comments.
>
> Response to questions:
>
> **1. Given that similar methods have applied LLMs for agent-based simulations with more advanced trading behaviors, could the authors clarify how MMARP’s contribution significantly advances this area?**
>
> Typically, the older works on LLM agents `[1, 2, 3]` utilize them as advisors to decide which stock to buy. These works are mainly focused on maximizing profits.
>
> Next, there are some recent works `[4, 5]` which use LLM agents to model individual investors, which is closer to our motivation. By doing so, companies or governments can then use these models to simulate the impact of new announcements or policies to see how they will affect the market. However, these works are usually small-scale simulations, given that one LLM can only simulate one single participant each time.
>
> Finally, MMARP seeks to simulate the *entire* market behavior when given some new information. This work validates the possibility of doing this through two main observations: One, the overall MMARP response behavior to different prices is similar to a demand curve, meaning that it can understand values the same way as humans. Two, our simulation allows us to make accurate price change forecasts, which shows the “reaction” from MMARP simulation to news information is very similar to the “reaction” from a real market. Also, most LLM agent works (including all the ones we cited) do not evaluate their performance on accuracy, only profitability.
>
> We would also like to note that most of the cited works are contemporaneous with our work (see the main rebuttal above), which highlights that this is a very new direction that is not widely explored yet.
>
> --
>
> `[1]` Yu, Yangyang, Haohang Li, Zhi Chen, Yuechen Jiang, Yang Li, Denghui Zhang, Rong Liu, Jordan W. Suchow, and Khaldoun Khashanah. “FinMem: A Performance-Enhanced LLM Trading Agent with Layered Memory and Character Design.” arXiv, December 3, 2023. https://doi.org/10.48550/arXiv.2311.13743.
>
> `[2]` Zhang, Wentao, Lingxuan Zhao, Haochong Xia, Shuo Sun, Jiaze Sun, Molei Qin, Xinyi Li, et al. “A Multimodal Foundation Agent for Financial Trading: Tool-Augmented, Diversified, and Generalist.” arXiv, June 28, 2024. https://doi.org/10.48550/arXiv.2402.18485.
>
> `[3]` Yu, Yangyang, Zhiyuan Yao, Haohang Li, Zhiyang Deng, Yupeng Cao, Zhi Chen, Jordan W. Suchow, et al. “FinCon: A Synthesized LLM Multi-Agent System with Conceptual Verbal Reinforcement for Enhanced Financial Decision Making.” arXiv, November 7, 2024. https://doi.org/10.48550/arXiv.2407.06567.
>
> `[4]` Gao, Shen, Yuntao Wen, Minghang Zhu, Jianing Wei, Yuhan Cheng, Qunzi Zhang, and Shuo Shang. “Simulating Financial Market via Large Language Model Based Agents.” arXiv, June 28, 2024. https://doi.org/10.48550/arXiv.2406.19966.
>
> `[5]` Zhang, Chong, Xinyi Liu, Zhongmou Zhang, Mingyu Jin, Lingyao Li, Zhenting Wang, Wenyue Hua, et al. “When AI Meets Finance (StockAgent): Large Language Model-Based Stock Trading in Simulated Real-World Environments.” arXiv, September 21, 2024. https://doi.org/10.48550/arXiv.2407.18957.

---

> ### Author Response · Authors · 2024-11-26
>
> **2. Could the authors provide theoretical support or evidence that demonstrates how collective responses from agents address or enhance LLMs' numerical reasoning?**
>
> In our updated version of the paper, we have redefined the challenges better, and provided theoretical support to show how repetitive prompting with the same prompts (which can be proxied by the generated weights) and prompting with a range of prices can help to simulate market behavior more precisely. This is largely to reduce sources of stochastic errors in both real-life and from the LLM’s unknown gaps in numerical understanding.

---

> > ### Author Response · Authors · 2024-11-26
> >
> > **3. Since all agents are based on the same LLM model with identical training data and input settings, how do the authors justify the use of demand-supply theory here?**
> >
> > The input settings are not completely identical. We make incremental changes, one at a time:
> >
> > First, we tune the prices provided, which shows that the higher the price, the less “quantity” or lower probability the LLM will do a Buy action. This is similar to the demand curve in economics theory, which show the possibility of using MMARP to simulate actual market response.
> >
> > Secondly, we then provide different context information, and we see that the plot adjusts accordingly based on the information (for example, plot shifts right when there is higher demand). This further shows that MMARP can simulate an actual market accurately, which is what we want to show.
> >
> > We take small steps to change each factor in the input prompt in order to verify each assumption is true before we move on, instead of making big changes just to show results improvement.

---

> > > ### Author Response · Authors · 2024-11-26
> > >
> > > **4. Would the authors expand to a larger dataset in multiple assets and different markets or including more diverse financial and general-purpose LLMs?**
> > >
> > > Following your comments, we have also added experimental results for two more market-related datasets, on Exchange Rates and Commodities. These are included in the updated version of the paper.

---

> > > > ### Comment · Reviewer_ATQ8 · 2024-11-27
> > > >
> > > > Thank you for the detailed response and clarifications. While I appreciate the effort to address the concerns raised, the key issues remain unresolved. The use of demand-supply theory is questionable, as all agents are based on the same LLM model with limited contextual variation, essentially representing a single client with consistent risk utility. This lack of agent heterogeneity undermines the applicability of demand curves and weakens the validity of the simulation from a financial theory perspective. Additionally, while the idea of simulating price formulation using LLM agents is interesting, it lacks grounding in established financial theories, and the oversimplified design makes it unrepresentative of real-world market behavior. The claims about improving numerical reasoning through repetitive prompting and price-range variations are also unconvincing, as there is insufficient evidence to demonstrate meaningful advancements in this area. Although the direction of the work is novel, these unresolved issues significantly limit its contributions. I appreciate the effort but will maintain my original score.

---

### Author Response · Authors · 2024-11-26

Dear reviewers, thank you all for your insightful comments.

Based on the comments, we have rewritten many parts of the paper to make our motivation and methods clearer, with theoretical justifications.

As a mitigating factor, we note that many of the cited works in the comments were very recent, and we did not consider them when doing our work (According to the ICLR reviewer guidelines `[1]`, works published within the last four months of DDL are considered contemporaneous. DDL was 1 October, and four months ago was 1 June).

For many agent-based works in Finance, the published dates are:

* **(28 June)** Simulating Financial Market via Large Language Model based Agents

* **(21 September)** When AI Meets Finance (StockAgent): Large Language Model-based Stock Trading in Simulated Real-world Environments

* **(7 November)** FINCON: A Synthesized LLM Multi-Agent System with Conceptual Verbal Reinforcement for Enhanced Financial Decision Making

Regardless, we have now included and compared with these works in our new version. The main difference between our work and most LLM financial agents works is that we are trying to do agent-based simulation over the whole financial market instead of simulating individual investors, and we evaluate based on the simulation’s forecasting ability, which is not commonly done (most works simply try to maximize profitability).

In addition, we want to highlight that the use of LLM in large-scale agent simulations is a very new topic. Only recently, we found some papers attempting to do it to simulate pandemic spread `[2]` **(23 October)** and election results `[3]` **(28 October)**. These papers are released almost a month after this conference’s DDL, which we could not have referred to. Also, this has also not been done in the Financial domain, which we do in our work.

We hope that reviewers will look at these points and considering revising our scores. Otherwise, we will also deeply appreciate it if we can get comments so that we can improve our work for the next conference. Thank you!

--

`[1]` [https://iclr.cc/Conferences/2025/ReviewerGuide](https://iclr.cc/Conferences/2025/ReviewerGuide)

`[2]`Chopra, Ayush, Shashank Kumar, Nurullah Giray-Kuru, Ramesh Raskar, and Arnau Quera-Bofarull. “On the Limits of Agency in Agent-Based Models.” arXiv, October 23, 2024. https://doi.org/10.48550/arXiv.2409.10568.

`[3]` Zhang, Xinnong, Jiayu Lin, Libo Sun, Weihong Qi, Yihang Yang, Yue Chen, Hanjia Lyu, et al. “ElectionSim: Massive Population Election Simulation Powered by Large Language Model Driven Agents.” arXiv, October 28, 2024. https://doi.org/10.48550/arXiv.2410.20746.

---

### Note · Authors · 2024-12-04

I have read and agree with the venue's withdrawal policy on behalf of myself and my co-authors.